# Early Supplemental Parenteral Nutrition in Critically Ill Children: An Update

**DOI:** 10.3390/jcm8060830

**Published:** 2019-06-11

**Authors:** An Jacobs, Ines Verlinden, Ilse Vanhorebeek, Greet Van den Berghe

**Affiliations:** Clinical Division and Laboratory of Intensive Care Medicine, Department of Cellular and Molecular Medicine, KU Leuven University Hospital, 3000 Leuven, Belgium; an.jacobs@kuleuven.be (A.J.); ines.verlinden@kuleuven.be (I.V.); ilse.vanhorebeek@kuleuven.be (I.V.)

**Keywords:** Pediatric Intensive Care Unit, enteral nutrition, early parenteral nutrition, critical illness

## Abstract

In critically ill children admitted to pediatric intensive care units (PICUs), enteral nutrition (EN) is often delayed due to gastrointestinal dysfunction or interrupted. Since a macronutrient deficit in these patients has been associated with adverse outcomes in observational studies, supplemental parenteral nutrition (PN) in PICUs has long been widely advised to meeting nutritional requirements. However, uncertainty of timing of initiation, optimal dose and composition of PN has led to a wide variation in previous guidelines and current clinical practices. The PEPaNIC (Early versus Late Parenteral Nutrition in the Pediatric ICU) randomized controlled trial recently showed that withholding PN in the first week in PICUs reduced incidence of new infections and accelerated recovery as compared with providing supplemental PN early (within 24 hours after PICU admission), irrespective of diagnosis, severity of illness, risk of malnutrition or age. The early withholding of amino acids in particular, which are powerful suppressors of intracellular quality control by autophagy, statistically explained this outcome benefit. Importantly, two years after PICU admission, not providing supplemental PN early in PICUs did not negatively affect mortality, growth or health status, and significantly improved neurocognitive development. These findings have an important impact on the recently issued guidelines for PN administration to critically ill children. In this review, we summarize the most recent literature that provides evidence on the implications for clinical practice with regard to the use of early supplemental PN in critically ill children.

## 1. Introduction

Optimal nutritional support is considered of paramount importance for critically ill children admitted to the pediatric intensive care unit (PICU), since malnutrition and inadequate nutrient delivery have been associated with worse clinical outcome [1,2]. Moreover, critically ill children have limited macronutrient stores and relatively higher energy requirements than adults admitted to the intensive care unit (ICU), which can lead to substantial caloric and macronutrient deficits [2,3,4]. The feeding is thought to attenuate the metabolic stress response, prevent oxidative cellular injury and modulate immune responses, and has led to a shift from nutritional support as adjunctive care to actual therapy of the critically ill child [5]. The enteral route is preferred for providing nutrition [5]. However, critically ill children are often too ill to be fed normally by mouth, and nasogastric or nasoduodenal tube feeding is often not tolerated because of gastric dysmotility or ileus. Interruption of enteral feeding also occurs frequently for various reasons, like medical or surgical contraindications, or radiology, bedside or surgical procedures [6]. Therefore, parenteral nutrition (PN) is often initiated to supplement the insufficient enteral intake. Nonetheless, official guidelines on timing and thresholds of initiation, composition and doses of supplemental PN vary widely [5,7,8,9]. Moreover, concerns about overfeeding have led to even more uncertainty [9]. A recent survey showed significant differences in nutritional practices in PICUs worldwide, in terms of macronutrient goals, estimation of energy requirements, timing of nutrient delivery and thresholds for starting supplemental PN [10]. In this review, we summarize the most recent literature findings affecting evidence and clinical practices with regard to the use of early PN in critically ill children. We searched PubMed up to April 2019, without language restrictions, using different combinations of the search terms “parenteral nutrition”, “PICU”, “early” and “pediatric critical illness”. We focused on publications of the last eight years, discussed in the context of earlier work. 

## 2. Timing of PN Initiation

Several observational studies have shown that malnutrition is associated with worse clinical outcome [1,2,11]. A macronutrient deficit has been associated with infections, weakness, prolonged mechanical ventilation and delayed recovery. For that reason, guidelines used to recommend that when provision of enteral nutrition (EN) is insufficient, impossible or contraindicated, supplemental PN should be initiated [7,12,13]. However, observational studies cannot assign causality to an association. Hence, the association between inadequate nutrition and worse clinical outcome might merely exist because of a non-optimal nutritional support for the sickest children, which are at the highest risk of adverse outcome. Although it seems intuitive that providing early nutrition will be beneficial, it does not necessarily mean that nutritional support in the early phase of critical illness will improve clinical outcome [9]. In critically ill adults, the large multicenter EPaNIC (Early versus Late Parenteral Nutrition in ICU, *n* = 4640) randomized controlled trial (RCT) showed that withholding supplemental PN until day eight of an ICU stay (late PN), and thus accepting a substantial macronutrient deficit, was associated with fewer ICU infections, a shorter duration of mechanical ventilation and renal-replacement therapy and a shorter ICU and total hospital stay as compared with initiating supplemental PN early (within 48 hours after ICU admission) [14]. Data generated by the broad international yearly survey of clinical nutrition practices “nutritionDay” revealed an important change in the pattern of PN prescription after publication of the EPaNIC results (Personal communication kindly shared by Prof. Dr. M. Hiesmayr, nutritionDay Project Leader). As compared with adults, critically ill children have limited stores of energy, fat and protein, as well as relatively higher energy requirements [2,3]. Since this makes them more vulnerable to a substantial caloric and macronutrient deficit, the effect of withholding supplemental PN in critically ill children could be different than in adults. Therefore, a multicenter PEPaNIC RCT (Early versus Late Parenteral Nutrition in the Pediatric ICU) was conducted [15] that investigated the same intervention in 1440 critically ill children aged 0–17 years in three PICUs in Belgium, the Netherlands and Canada. Withholding supplemental PN during the first week in critically ill children resulted in fewer new infections, a shorter dependency on mechanical ventilation and general intensive care and a shorter hospital stay as compared with providing PN early (within 24 hours after PICU admission; see Figure 1). The clinical superiority of late PN was more pronounced in children than it was in adults, and was shown irrespective of diagnosis, severity of illness, risk of malnutrition or age of the child [15]. This last finding was surprising, since neonates are more susceptible to macronutrient deficits than older children [7], raising concerns by experts [16,17,18]. To address these concerns, a secondary analysis of the PEPaNIC trial was performed to investigate the effects of withholding PN for one week in 209 critically ill neonates who did not (or could hardly) tolerate EN [19]. Analyses were performed for term neonates aged up to four weeks, up to one week and younger than one day. Late PN resulted in fewer nosocomial infections in neonates aged up to one week and younger than one day, and in shorter dependency on intensive care and mechanical ventilation for all studied age groups of neonates. Hence, term neonates also benefited from withholding PN during the first week in the PICU, which is in agreement with findings for older children and adults [14,15]. Moreover, there was a more pronounced benefit of late PN in the youngest children, as shown in Figure 1. Since a macronutrient deficiency is presumed to be more detrimental during acute illness in undernourished children [5], a second subanalysis of the PEPaNIC RCT was performed, investigating the effects of withholding supplemental PN during the first PICU week in a subgroup of critically ill children who were undernourished upon admission to the PICU [20]. Undernourishment was defined as a weight-for-age z score lower than −2 in children younger than one year, and a body mass index-for-age z score lower than −2 in children one year or older. This identified 289 of 1440 PEPaNIC patients (20%) with undernourishment upon PICU admission. Among the undernourished patients, late PN reduced the absolute risk of new infections and shortened the duration of PICU stay. These effect sizes of late PN were even larger than in the main trial cohort of the PEPaNIC RCT. Late PN did not affect the safety outcomes of mortality, incidence of hypoglycemia or weight deterioration during PICU stay in the undernourished patients. A larger longitudinal study of all PEPaNIC patients with weight z scores available on admission and on the last day in the PICU showed that weight deterioration during PICU stay was associated with worse clinical outcomes, but that withholding supplemental PN during the first week did not alter weight z score deterioration during the PICU stay [21]. 

The benefits of withholding supplemental PN during the first week in the PICU appeared not only present from a clinical point of view, but also from a health–economic perspective. A cost-effectiveness study indeed showed that the total direct medical costs were considerably lower with late PN as compared with early supplemental PN initiation [22]. This cost saving was beyond the expected lower costs for the use of PN itself, since avoidance of new infections by late PN yielded the largest cost reduction. 

A possible limitation of the PEPaNIC RCT is the use of standard equations for the estimation of energy requirements instead of indirect calorimetry [23]. However, the use of indirect calorimetry for estimating energy expenditure does not seem to be accurate [24], or feasible [25], and is not frequently used in daily practice [10,26]. 

Apart from the PEPaNIC RCT, no other randomized controlled trial investigating the use or timing of supplemental PN in critically ill children has been published in the last eight years. A limited number of observational studies on the use of supplemental PN and over- and underfeeding in PICUs showed different results [2,24,27]. A retrospective single center study showed that late initiation of supplemental PN was associated with a higher nosocomial infection rate as compared with early initiation of supplemental PN [27]. In contrast, an observational study in 31 PICUs showed that the use of PN in general was associated with higher mortality [2]. Another retrospective study determining the incidence of over- and underfeeding in 139 children admitted to a tertiary PICU showed that underfeeding was associated with shorter durations of PICU and hospital stays, as well as with fewer ventilation days, as compared with appropriately fed and overfed patients [24]. However, the observational design of these studies holds a risk of bias by confounding variables, especially in nutritional research [28]. Therefore, comparison with the results of the PEPaNIC RCT is challenging. Further randomized controlled trials are warranted to determine the ideal time point for initiation of supplemental PN in PICUs.

## 3. Early PN Composition and the Role of Macronutrients

Although extensive guidelines on the composition of PN in critically ill children are available [5], a recent survey on nutritional practices in PICUs worldwide showed a wide variation in parenterally administered doses of protein, lipids and glucose [10]. Protein targets in particular seem to be a point of discussion. Several studies have assessed the association between protein delivery to and clinical outcome of critically ill children [2,29,30]. In an observational international cohort study that included 500 critically ill children, mortality at 60 days was higher in patients who received PN independently of the amount of energy or protein intake [2]. However, an important severity of illness bias has to be taken into account, since patients who are less sick are more likely to better tolerate EN. The study adjusted for severity of illness using admission scores, but data for calculating this severity of illness score were missing in 31% of the included patients, and the choice of severity of illness score differed between the participating centers [2]. Another large multicenter observational study of the same group showed an association between higher enteral protein intake and lower odds of mortality in more than 1200 mechanically ventilated critically ill children [29]. The effect was dose-dependent, and independent of energy intake. Again, the incomplete datasets and the lack of uniform usage for severity of illness scores, and the substantial number of patients who received EN, could potentially bias these observations. The authors reasoned that an increased demand in amino acids in catabolic disease, such as critical illness, could contribute to increased higher protein degradation from muscle to ensure bodily functions [29], which is associated with poor outcome [31]. By providing proteins, the synthesis of muscle proteins might be boosted, and thereby muscle loss could be prevented, possibly limiting the severity of intensive care unit-acquired weakness [32]. Nevertheless, a preplanned secondary analysis of the adult EPaNIC study did not support this concept, as increased macronutrient intake with early PN, including more amino acids, did not counteract muscle atrophy and actually increased the risk of developing clinically relevant muscle weakness in the ICU [33]. Interestingly, in a preplanned secondary analysis of the PEPaNIC RCT, the dose of amino acids was actually associated with more infections and a longer dependency on mechanical ventilation and other intensive medical care in children admitted to the PICU [30]. This risk of harm associated with early amino acid administration was elevated even at low doses of administered amino acids. A possible explanation for the difference between these results and the ones from the previously cited observational studies is the randomized design of the PEPaNIC trial, in which the doses of macronutrients differed from patient to patient and ranged widely [30]. In critically ill adults, three RCTs could not show benefits from early amino acid supplementation [34,35,36,37], but clinical trials on the effects of amino acid administration on clinical outcome in critically ill children in a randomized manner are lacking [38]. In contrast with the harm of amino acid administration, the secondary analysis of the PEPaNIC RCT suggested a benefit of glucose and lipid administration. Indeed, administering more glucose during the first three days of PICU stay was independently associated with fewer infections, and administering more lipids was independently associated with earlier PICU discharge [30]. Clearly, large-scale prospective RCTs in critically ill children are needed to identify the optimal composition of supplemental PN [9,38]. 

## 4. Impact of Early PN on Long-Term Outcome 

Because of new insights in diagnostic and therapeutic measures in the field of pediatric critical care medicine [39], centralization of care [40] and specialized staff training and education [41,42], there has been an important decline in the mortality rate in PICUs over the last decades [43,44]. However, this improved survival has led towards a shift to considerable long-term morbidity, years after discharge [45,46]. This has been most thoroughly documented in regards to impairment of neurocognitive development, but it also includes growth retardation and may comprise poor physical functioning and reduced quality of life [47,48,49,50]. The fact that children are treated in the PICU during crucial developmental phases likely plays a role. Interestingly, it appears that, to a certain extent, neurocognitive outcome is modifiable, as shown by the attenuation of neurocognitive impairment with the prevention of hyperglycemia during intensive care [51]. Treatments in PICUs that have been shown to cause neurodevelopmental harm, such as anesthetic and analgesic agents [52,53] and toxicants such as phthalates that leach from indwelling medical devices [54], may also be targets for research into safer alternatives. Concerning nutrition in PICUs, in relation to long-term outcome, experts were concerned about the safety of withholding early supplemental PN in neonates in view of the more frequent episodes of hypoglycemia observed in the late PN arm of the PEPaNIC RCT [15,17]. However, in a previous large randomized controlled trial investigating the effect of tight glycemic control on morbidity and mortality in PICUs and on long-term neurocognitive development, a high incidence of brief hypoglycemia with tight glycemic control was not associated with harm to neurocognitive development, as documented four years later [51]. The proportion of neonates included in the PEPaNIC RCT was similar to that in the tight glycemic control trial [15,55]. In a preplanned two-year follow-up study, in which all patients included in the PEPaNIC RCT were approached for possible assessment of physical and neurocognitive development, exposure to hypoglycemia also did not associate with the investigated long-term outcomes [53]. Moreover, the main results of this follow-up study showed no adverse effects of withholding supplemental PN during the first week in PICUs on survival, anthropometrics, health status and neurocognitive development. In fact, omitting early supplemental PN in PICUs improved parent-reported executive functioning (inhibition, working memory, metacognition and overall executive functioning), externalizing behavioral problems and visual-motor integration two years later, as compared with early supplemental PN. In particular, a better inhibitory control was observed (Figure 2). Since poor inhibitory control in children contributes to impulsive and destructive behaviors that upset or harm others [56], delaying supplemental PN can have important consequences on daily life and social environments later in life. The long-term effects of late versus early supplemental PN were more pronounced in patients who were younger than one year of age at the time of PICU admission as compared with older children. This age-dependent vulnerability supports the hypothesis that the harm induced by early supplemental PN might be caused by a direct metabolic insult to the developing brain, since it was not statistically explained by the acute effects of the intervention itself, such as the increased incidence of new infections or delayed recovery. However, further research is warranted to unravel the underlying mechanisms that would provide support for this hypothesis. Although long-term outcomes and quality-of-life years after PICU discharge have gained great importance in research [45,46], investigating these outcomes is logistically challenging, expensive and time-consuming. To investigate whether these long-term effects persist or change over time, a four-year follow-up study of the PEPaNIC RCT is currently ongoing, of which the results are expected by 2020. 

## 5. Potential Mechanisms Underlying Harm by Early PN

Several mechanisms may contribute to an adverse clinical impact of administering PN in the early phase of critical illness in children. As early PN has affected both short-term and long-term outcomes of patients, carryover effects persisting in the long term must play a role. 

### 5.1. Potential Mechanisms Underlying the Adverse Impact of Early PN on Short-Term Outcome

As discussed, it was the administration of more amino acids that was associated with adverse short-term outcomes evoked by early PN in the PEPaNIC study [30]. The mechanisms underlying the potential harm of amino acids in this context remain to be unraveled, but it is possible to speculate. First, amino acids are powerful suppressors of autophagy [57], a pathway shown to be crucial in critically ill humans and animals for innate immunity and the removal of cellular damage [58,59]. Autophagy activation appeared suppressed in muscle from randomized adult patients exposed to early PN as compared with late PN in the EPaNIC trial, and amino acids in particular suppressed autophagy activation in a rabbit model of prolonged critical illness [33,58,59]. Second, macronutrients and amino acids in particular have long been assumed to counteract the hypercatabolic response to critical illness, which mobilizes amino acids by (mainly muscle) protein breakdown to guarantee substrate delivery to vital tissues [60]. More specifically, administration of exogenous amino acids was thought to circumvent the need for endogenous amino acid release and to stimulate muscle protein synthesis. However, once amino acid doses exceed anabolic capacity, they no longer stimulate muscle protein synthesis but are shuttled towards the liver for production of urea [61]. Such increased plasma urea concentrations were observed in early PN patients in the PEPaNIC RCT as compared with late PN patients [15]. In critically ill adults who received early PN, the administered amino acids did not counteract muscle wasting [33], but increased plasma urea and urinary nitrogen excretion (with a net waste of 63% of the extra nitrogen intake from early PN in the first two weeks) [62], which can cause harm to both liver and kidney. These results are in line with the EAT-ICU trial (Early goal-directed nutrition in ICU patients) results, in which increased urea production—but no beneficial effect on outcome—was seen with early goal-directed nutrition versus standard nutritional care in critically ill adults [63]. Altogether, these findings suggest that muscle wasting during the acute phase of critical illness may be considered an adaptive response designed to provide substrates for gluconeogenesis in order to meet the energy requirements of vital organs [32].

### 5.2. Potential Mechanisms Underlying the Adverse Impact of Early PN on Long-Term Outcome

Underlying mechanisms of the long-term harm caused by early PN in the context of critical illness remain largely unraveled. The clinical benefits of late PN observed far beyond the intervention window suggests that early PN induces carry-over “memory” effects with a negative impact on long-term outcome. Poor long-term outcomes in other conditions have been related to accelerated telomere shortening and the induction of aberrant so-called “epigenetic” changes. Importantly, inadequate nutrition may cause both adverse effects. Hence, these processes may also play a role in the developmental impairment of critically ill children and the adverse impact of early PN on neurocognitive development. 

Telomeres are nucleoprotein complexes at the end of human chromosomes that shorten with each cell cycle. Telomere shortening can be accelerated by environmental and lifestyle factors [64,65], including excessive food consumption and/or unhealthy nutrition [66,67]. It has been demonstrated that critically ill children enter the PICU with significantly shorter leukocyte telomeres than matched healthy children [68]. More importantly, early PN had a telomere-shortening effect as compared with late PN in critically ill children between PICU admission and discharge, independent of baseline risk factors and post-randomization factors. Whether this accelerated telomere shortening contributes to long-term developmental impairment, and particularly the neurocognitive impairment caused by early PN, remains to be investigated. 

The involvement of aberrant epigenetic changes in long-term consequences after acute events in life appears plausible. Epigenetics refers to the study of heritable changes in gene expression that do not involve changes in the underlying DNA sequence. Epigenetic changes play an important role in physical and neurocognitive development [69,70,71,72]. The most stable epigenetic change is the methylation or de-methylation of DNA. This is the attachment to or removal of a methyl group from a nucleotide, which occurs almost exclusively at the 5’ carbon in the cytosine residue of a CpG dinucleotide [69,70]. Alterations in DNA methylation have been implicated in the adverse effects of various environmental stressors, such as inadequate nutrition (both undernutrition and overfeeding), that have been shown to impact long-term health and disease [73]. Particularly during early life, DNA methylation changes may bring about long-term effects [71,72,74]. Data are needed on whether nutritional management in the early phase of critical illness induces aberrant changes in DNA methylation, which could explain the adverse impact of early PN on neurocognitive development. 

## 6. Conclusions and Newest Guidelines

Nutritional support is considered to be an important aspect of patient care in PICUs [26], with large differences in PICUs worldwide regarding practices of supplemental PN for patients in whom EN as the preferred route is insufficient or impossible [5,10]. A large multicenter RCT showed that withholding supplemental PN throughout the first week in the PICU was clinically superior for short-term outcome as compared with initiating supplemental PN within 24 hours after admission [19]. This was the case independent of age [19] or nutritional status [20]. The administered amino acids in particular appeared to explain the adverse impact of early provision of PN, possibly via suppression of required autophagy activation and the shuttling of amino acids to urea production with harmful effects to the liver and kidney. In the long-term, omitting PN during the first week of pediatric critical illness did not show harm, and actually improved executive functioning, behavioral problems and visual-motor integration, as compared with administering PN early. These findings had an important impact on recent ESPGHAN (European Society for Pediatric Gastroenterology Hepatology and Nutrition) pediatric PN guidelines [60], in which it is advised to consider withholding PN, including amino acids, for one week in critically ill infants, children and adolescents. However, the lack of other RCTs in this specific field makes it challenging to compare these findings with other available recent studies. Further research, in the form of multicenter RCTs, is warranted to determine the optimal composition and ideal timing of initiation of supplemental PN in critically ill children. 

## Figures and Tables

**Figure 1 jcm-08-00830-f001:**
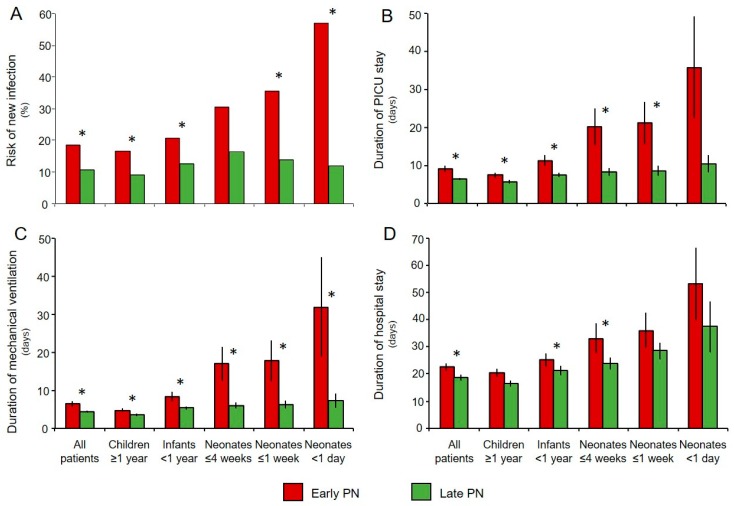
Bars represent incidence of (**A**) the risk of acquiring a new infection in PICUs (percentage), (**B**) duration (days) of PICU stay, (**C**) duration (days) of mechanical ventilation, and (**D**) duration (days) of total hospital stay. Whiskers indicate standard errors of the mean. Asterisks represent *p* values < 0.05 obtained with multivariable analysis adjusting for baseline risk factors (treatment center, age, risk of malnutrition (STRONGkids score), diagnosis upon admission and severity of illness (Pediatric Logistic Organ Dysfunction (PeLOD) score, and Pediatric Risk of Mortality 2 (PIM2) score) for all patients; treatment center, risk of malnutrition, diagnosis upon admission and severity of illness for children and infants; and treatment center, type of illness upon admission (medical, surgical cardiac, surgical other), severity of illness and weight for age z score for neonates).

**Figure 2 jcm-08-00830-f002:**
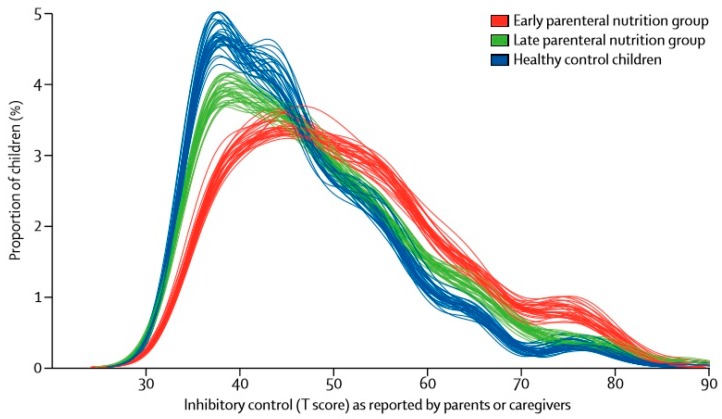
Density estimates for inhibitory function as reported by parents or caregivers. Densities, which correspond to the proportions of children with a certain score (equivalent to a smoothed histogram), are shown separately for healthy control children and for PEPaNIC participants who were randomly assigned to receive late PN or early PN. Higher scores indicate worse functioning. Each line corresponds to one of 31 imputed datasets. Figure reprinted from [53] with permission from Elsevier.

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
