# Peer review of "Early Supplemental Parenteral Nutrition in Critically Ill Children: An Update"

_jcm, 2019, doi:10.3390/jcm8060830_

Reviewer 1 Report

The reviewed paper is a description and commentary of the authors' previous articles. Broadly discussed are the following papers, and some sentences even written out of them:

15. Fivez, T.; Kerklaan, D.; Mesotten, D.; Verbruggen, S.; Wouters, P.J.; Vanhorebeek, I.; Debaveye, Y.; 353 Vlasselaers, D.; Desmet, L.; Casaer, M.P., et al. Early versus Late Parenteral Nutrition in Critically Ill 354 Children. N Engl J Med 2016, 374, 1111-1122, doi:10.1056/NEJMoa1514762.

19. van Puffelen, E.; Vanhorebeek, I.; Joosten, K.F.M.; Wouters, P.J.; Van den Berghe, G.; Verbruggen, S. Early 362 versus late parenteral nutrition in critically ill, term neonates: a preplanned secondary subgroup analysis 363 of the PEPaNIC multicentre, randomised controlled trial. Lancet Child Adolesc Health 2018, 2, 505-515, 364 doi:10.1016/S2352-4642(18)30131-7.

21. van Puffelen, E.; Hulst, J.M.; Vanhorebeek, I.; Dulfer, K.; Van den Berghe, G.; Verbruggen, S.C.A.T.; 371 Joosten, K.F.M. Outcomes of Delaying Parenteral Nutrition for 1 Week vs Initiation Within 24 Hours 372 Among Undernourished Children in Pediatric Intensive Care: A Subanalysis of the PEPaNIC 373 Randomized Clinical Trial. JAMA Netw Open 2018, 1, e182668, doi:10.1001/jamanetworkopen.2018.2668. 374

22. van Puffelen, E.; Hulst, J.M.; Vanhorebeek, I.; Dulfer, K.; Van den Berghe, G.; Joosten, K.F.M.; 375 Verbruggen, S. Effect of late versus early initiation of parenteral nutrition on weight deterioration during 376 PICU stay: Secondary analysis of the PEPaNIC randomise

26. Vanhorebeek, I.; Verbruggen, S.; Casaer, M.P.; Gunst, J.; Wouters, P.J.; Hanot, J.; Guerra, G.G.; 389 Vlasselaers, D.; Joosten, K.; Van den Berghe, G. Effect of early supplemental parenteral nutrition in the 390 paediatric ICU: a preplanned observational study of post-randomisation treatments in the PEPaNIC trial. 391 Lancet Respir Med 2017, 5, 475-483, doi:10.1016/S2213-2600(17)30186-8.

49. Verstraete, S.; Verbruggen, S.C.; Hordijk, J.A.; Vanhorebeek, I.; Dulfer, K.; Guiza, F.; van Puffelen, E.; 454 Jacobs, A.; Leys, S.; Durt, A., et al. Long-term developmental effects of withholding parenteral nutrition 455 for 1 week in the paediatric intensive care unit: a 2-year follow-up of the PEPaNIC international, 456 randomised, controlled trial. Lancet Respir Med 2018, 10.1016/S2213-2600(18)30334-5, doi:10.1016/S2213-457 2600(18)30334-5.

If the presented work aims to update the information on parenteral nutrition of critically ill children, then it would also be necessary to take up other topics than those dealt with by the authors, e.g. the effect of the type of amino acids and fat emulsion on clinical results, immunomodulatory nutrition, aluminum toxicity, etc.

In this form, the reviewed article is only a summary of the research carried out by the authors with a small commentary and discussion with other articles. There is also no detailed discussion of the authors' widely discussed publications with existing guidelines.

Additional remarks:

- In the title there is “update” and many references are before 2010  

- the authors cite non-published data (line 72)

- line 66 should be ICU not PICU

- lines 99-108: z sore, z-score or Z-score - should be unified

- fig 1 - what is the total energy graph shown for? There is no information about the source of this figure

Author Response

Reviewer 1.

The reviewed paper is a description and commentary of the authors' previous articles. Broadly discussed are the following papers, and some sentences even written out of them:

15. Fivez, T.; Kerklaan, D.; Mesotten, D.; Verbruggen, S.; Wouters, P.J.; Vanhorebeek, I.; Debaveye, Y.; 353 Vlasselaers, D.; Desmet, L.; Casaer, M.P., et al. Early versus Late Parenteral Nutrition in Critically Ill 354 Children. N Engl J Med 2016, 374, 1111-1122, doi:10.1056/NEJMoa1514762.

19. van Puffelen, E.; Vanhorebeek, I.; Joosten, K.F.M.; Wouters, P.J.; Van den Berghe, G.; Verbruggen, S. Early 362 versus late parenteral nutrition in critically ill, term neonates: a preplanned secondary subgroup analysis 363 of the PEPaNIC multicentre, randomised controlled trial. Lancet Child Adolesc Health 2018, 2, 505-515, 364 doi:10.1016/S2352-4642(18)30131-7.

21. van Puffelen, E.; Hulst, J.M.; Vanhorebeek, I.; Dulfer, K.; Van den Berghe, G.; Verbruggen, S.C.A.T.; 371 Joosten, K.F.M. Outcomes of Delaying Parenteral Nutrition for 1 Week vs Initiation Within 24 Hours 372 Among Undernourished Children in Pediatric Intensive Care: A Subanalysis of the PEPaNIC 373 Randomized Clinical Trial. JAMA Netw Open 2018, 1, e182668, doi:10.1001/jamanetworkopen.2018.2668. 374

22. van Puffelen, E.; Hulst, J.M.; Vanhorebeek, I.; Dulfer, K.; Van den Berghe, G.; Joosten, K.F.M.; 375 Verbruggen, S. Effect of late versus early initiation of parenteral nutrition on weight deterioration during 376 PICU stay: Secondary analysis of the PEPaNIC randomise

26. Vanhorebeek, I.; Verbruggen, S.; Casaer, M.P.; Gunst, J.; Wouters, P.J.; Hanot, J.; Guerra, G.G.; 389 Vlasselaers, D.; Joosten, K.; Van den Berghe, G. Effect of early supplemental parenteral nutrition in the 390 paediatric ICU: a preplanned observational study of post-randomisation treatments in the PEPaNIC trial. 391 Lancet Respir Med 2017, 5, 475-483, doi:10.1016/S2213-2600(17)30186-8.

49. Verstraete, S.; Verbruggen, S.C.; Hordijk, J.A.; Vanhorebeek, I.; Dulfer, K.; Guiza, F.; van Puffelen, E.; 454 Jacobs, A.; Leys, S.; Durt, A., et al. Long-term developmental effects of withholding parenteral nutrition 455 for 1 week in the paediatric intensive care unit: a 2-year follow-up of the PEPaNIC international, 456 randomised, controlled trial. Lancet Respir Med 2018, 10.1016/S2213-2600(18)30334-5, doi:10.1016/S2213-457 2600(18)30334-5.

If the presented work aims to update the information on parenteral nutrition of critically ill children, then it would also be necessary to take up other topics than those dealt with by the authors, e.g. the effect of the type of amino acids and fat emulsion on clinical results, immunomodulatory nutrition, aluminum toxicity, etc.

Reply: We would like to thank the reviewer for the time invested in this review. The objective of this article was to discuss most recent literature findings affecting evidence and clinical practices with regard to the use of early supplemental PN in critically ill children. We agree that the “early” aspect of supplemental PN in PICU might not have been very clear. We therefore have now clarified this in the manuscript’s title, subtitles and statement on the objective in the abstract and introduction. In the introduction, we have also clarified the search we performed for collecting the literature findings. The end of the introduction now reads as follows:

“In this review, we summarize the most recent literature findings affecting evidence and clinical practices with regard to the use of early PN in critically ill children. We searched PubMed up to April, 2019, without language restrictions, with different combinations of the search terms “parenteral nutrition”, “PICU”, “early”, pediatric critical illness”. We focused on publications of the last 8 years, discussed in the context of earlier work.”

In the last 8 years, other studies on the subject found with this search were scarce and observational. The non-randomized design of these studies holds risk of bias by confounding variables – in particular when investigating the effect of nutritional management. Thus, their level of evidence is rather poor. We have now added a section discussing these studies at the end of “2. Timing of PN initiation”:

“Apart from the PEPaNIC RCT, no other randomized controlled trial investigating the use or timing of supplemental PN in critically ill children has been published in the last 8 years. A limited number of observational studies on the use of supplemental PN and over- and underfeeding in PICU showed different results [25,28]. A retrospective single center study showed that late initiation of supplemental PN was associated with a higher nosocomial infection rate as compared with early initiation of supplemental PN [28]. In contrast, an observational study in 31 PICUs showed that the use of PN in general was associated with higher mortality [2]. Another retrospective study determining the incidence of over- and under-feeding in 139 children admitted to a tertiary PICU, showed that underfeeding was associated with shorter duration of PICU and hospital stay, as well as with fewer ventilation days, as compared with appropriately fed and overfed patients [25]. However, the observational design of these studies holds risk of bias by confounding variables, especially in nutritional research [29]. Therefore, comparison with the results of the PEPaNIC RCT is challenging. Further randomized controlled trials are warranted to determine the ideal time point for initiation of supplemental PN in the PICU.”

In this form, the reviewed article is only a summary of the research carried out by the authors with a small commentary and discussion with other articles. There is also no detailed discussion of the authors' widely discussed publications with existing guidelines.

Reply: As stated above in the reply on the previous comment, we have now added a section discussing the recent observational studies on the subject. As mentioned, qualitative research on this subject is scarce. Since the PEPaNIC trial is the only randomized controlled trial investigating early supplemental PN in critically ill children, we believe that its thorough discussion is required in this review. We assume that this provides an explanation for the fact that our group had received the invitation to write this review.  

However, we have now added a possible limitation of the PEPaNIC RCT to the paragraph:

“A possible limitation of the PEPaNIC RCT is the use of standard equations for the estimation of energy requirements, instead of indirect calorimetry [24]. However, the use of indirect calorimetry for estimating energy expenditure does not seem to be accurate [25], or feasible [26], and is not frequently used in daily practice [10,27].”

Additional remarks:

- In the title there is “update” and many references are before 2010 

Reply: As stated above, we have now specified the search that we performed for this review and we have added reference to more recent observational studies in “2. Timing of PN initiation”. Furthermore, we would like to emphasize that the references from before 2010 are meant to delineate the context of the more recent studies.

- the authors cite non-published data (line 72)

Reply: As stated in the manuscript, this indeed refers to unpublished data that were shared with permission of the author. Although not published yet, we believe that these data are an important contribution to the update on the research on the use of early supplemental PN.

- line 66 should be ICU not PICU

Reply: Adapted as requested.

- lines 99-108: z sore, z-score or Z-score - should be unified

Reply: Adapted as requested. We have now used a uniform spelling for “z score”. 

- fig 1 - what is the total energy graph shown for? There is no information about the source of this figure

Reply: We have now adapted Figure 1 and have removed the total energy graphs. We have now summarized in one illustration the increasing benefit of late PN with younger ages, by showing the differences in outcomes (incidence of the acquisition of a new infection in PICU, duration of mechanical ventilation, PICU stay and total hospital stay) between early PN and late PN in different age groups of the PEPaNIC RCT and emphasized this in the text.

“Hence, also term neonates benefited from withholding PN during the first week in PICU, in agreement with findings for older children and adults [14,15]. Moreover, there is a more pronounced benefit of late PN in the youngest children, as shown by Figure 1.”

Reviewer 2 Report

Thank you for submitting the manuscript “supplemental parenteral nutrition in critically ill children” an update, with the aim to summarize recent evidence and clinical practices on the initiation and use of PN in critically children.

A review of current evidence is needed, but available high quality studies are scarce.  In general you refer broadly to early and late PN, with most references from the EPaNIC and PEPaNIC trial. Both these studies are important and are well highlighted. The role of macronutrients and the impact of PN on long-term outcome are also described.

However, the limitations of the PEPaNIC trial are not discussed in detail in the submitted manuscript. Early initiation of PN is not defined and might mislead the reader to think that starting PN after 3-5 days has the same negative effects as starting within 24 hours after admission to the PICU, something which has not been studied. The limitations of the PEPaNIC trial and the meaning of early PN in this context needs to be better clarified throughout the manuscript, for instance by a more proper definition of early PN: At PICU admission or within 24 hours after admission to the PICU. The lack of studies within the field, makes such a review difficult, and it is itherefore also mportant to describe the limitations of the conclusions drawn, and to be more humble.

Specific comments:

Abstract:

Line 24, early PN should be better defined (as mentioned above)

The purpuse of the paper is not presented, just outcomes from the PEPaNIC trial. I suggest that the last part is rewritten.

Introduction:

Line 38: Does the references sited support that critically ill children have limited macronutrient stores as compared to critically ill adults, or is it just the higher energy requirements that result in relative lower stores? If so, please rephrase.

Line 46-47: The indication for PN could be more extensive described

Line 52:.. most recent literature findings… What does this encompass? Time aspect? What kind of search?

Timing of PN initiation:

Line 57-59: “Therefore, guidelines used to recommend that when provision of enteral nutrition (EN) is insufficient, impossible or contra-indicated, supplemental PN should be initiated soon after PICU admission (ref 7,12,13).

To my knowledge not any of these references have advocated PN soon after PICU admission, ref 7: In small preterm infants, PN must be instituted shortly after birth, in older children and in adolescence up to 7 days may be tolerated, depending on age, nutritional status, and the disease, surgery or medical intervention. Ref 12: No evidence for or against the need for nutritional support during the first week of life (Conventional feeding regarded as after 48 h). Ref. 13: Current practice is the initiation of enteral feeds within 48-72 h after admission to a PICU, and PN should only be considered if EN is not possible.

The authors need to find other references or rephrase.

Line 69: early PN (please specify)

Line 73-74: This is the same sentence as line 38, and same comment

Line 106-109: This is also from the PEPaNIC trial and this could be clearer in the text

Line 119-122: Only other trial sited. The authors do not discuss the differences between the findings in this trial and the findings in the PEPaNIC trial. This could be elaborated.

PN and the role of macronutrients

Line 159 and 160: The possible explanation seems a bit odd. Most observational studies a less stringent than RCT, thus with more diversity. RCT are defined, and will often have less differences between the included subjects. I do not know if it is correct to say that it was the RCT design, but rather the lack of restrictions on the participating centers, so that the nutritional intake (apart from when to start PN), had an observational character…

Impact of PN on long-term outcome

According to this review, parents reported a better inhibitory control among children in the late PN group of the PEPaNIC trial after 2 years.

Line 200-204 “Delaying supplemental PN has important consequences for daily life and the social environment later in life” …..  and further … “ supports the hypothesis that the harm induced by early supplemental PN might be caused by a direct metabolic insult….”

The authors are very conclusive. These limitations of the PEPaNIC trial does not allow such firm conclusions.

Conclusions and newest guidelines

Line 287: …”compared with initiating supplementa PN early”, please specify

Include a paragraph on the lack of studies found for this review and the limitations for the conclusions.

Line 295- 297: “Further research, in the form of multicenter RCTs, is warranted to determine the optimal composition and timing supplemental PN beyond the first week of critical illness in children.

Please rephrase, we still do not know if PN started at 3, 4, 5 or 6 days are inferior to withholding PN for a week.

Author Response

Reviewer 2

Thank you for submitting the manuscript “supplemental parenteral nutrition in critically ill children” an update, with the aim to summarize recent evidence and clinical practices on the initiation and use of PN in critically children.

A review of current evidence is needed, but available high quality studies are scarce.  In general you refer broadly to early and late PN, with most references from the EPaNIC and PEPaNIC trial. Both these studies are important and are well highlighted. The role of macronutrients and the impact of PN on long-term outcome are also described.

However, the limitations of the PEPaNIC trial are not discussed in detail in the submitted manuscript. Early initiation of PN is not defined and might mislead the reader to think that starting PN after 3-5 days has the same negative effects as starting within 24 hours after admission to the PICU, something which has not been studied. The limitations of the PEPaNIC trial and the meaning of early PN in this context needs to be better clarified throughout the manuscript, for instance by a more proper definition of early PN: At PICU admission or within 24 hours after admission to the PICU. The lack of studies within the field, makes such a review difficult, and it is itherefore also mportant to describe the limitations of the conclusions drawn, and to be more humble.

Reply: We would first like to thank the reviewer for the time and effort invested in this review.

We have now clearly defined “early PN” by adding “within 24 hours after PICU admission” in the abstract. We have now also clarified this in “2. Timing of PN initiation”, for EPaNIC and PEPaNIC respectively. 

As requested, we have now discussed the use of standard equations instead of indirect calorimetry as a possible limitation of the PEPaNIC RCT. We have added this section at the end of the paragraph describing the PEPaNIC RCT and its secondary analyses:

A possible limitation of the PEPaNIC RCT is the use of standard equations for the estimation of energy requirements, instead of indirect calorimetry [24]. However, the use of indirect calorimetry for estimating energy expenditure does not seem to be accurate [25], or feasible [26], and is not frequently used in daily practice [10,27].”

In the conclusion, we have now added that the lack of other RCTs on this subject is challenging for comparison with other available studies. We have also rephrased the last sentence of the conclusion to avoid confusion as mentioned by the reviewer. This section now reads as follows:

“However, the lack of other RCTs in this specific field makes it challenging to compare these findings with other available recent studies. Further research, in the form of multicenter RCTs, is warranted to determine the optimal composition and ideal timing of initiation of supplemental PN in critically ill children.”

Although we acknowledge that the PEPaNIC RCT is indeed the only recent study providing high level of evidence within this subject, we hope to have provided a more nuanced approach in the replies on the specific comments below and in the corresponding adaptations in the manuscript, as requested by the reviewer.  

Specific comments:

Abstract:

Line 24, early PN should be better defined (as mentioned above)

Reply: Adapted as requested. This sentence now reads:

“The PEPaNIC randomized controlled trial recently showed that withholding PN in the first week in PICU reduced the incidence of new infections and accelerated recovery as compared with providing supplemental PN early (within 24 hours after PICU admission), irrespective of diagnosis, severity of illness, risk of malnutrition or age.”

The purpuse of the paper is not presented, just outcomes from the PEPaNIC trial. I suggest that the last part is rewritten.

Reply: We have added a description of the purpose of the review at the end of the abstract.

“In this review, we summarize the most recent literature that provided evidence with implications for clinical practice with regard to the use of early supplemental PN in critically ill children.”

In addition, since the objective focuses on the use of early supplemental PN, we have now changed the main title and one of the subtitles to:

Early supplemental parenteral nutrition in critically ill children: an update”

“3.         Early PN composition and the role of macronutrients”

“4.         Impact of early PN on long-term outcome”

Introduction:

Line 38: Does the references sited support that critically ill children have limited macronutrient stores as compared to critically ill adults, or is it just the higher energy requirements that result in relative lower stores? If so, please rephrase.

Reply: As compared with critically ill adults, critically ill children both have limited macronutrient stores and relatively higher energy requirements. The limited stores are therefore not only due to the higher requirements. These differences with adults are not absolute, so we have now adapted this sentence, which now reads as follows:

“Moreover, critically ill children have limited macronutrient stores and relatively higher energy requirements than adults admitted to the ICU, which can lead to substantial caloric and macronutrient deficits.”

Line 46-47: The indication for PN could be more extensive described

Reply: We are not quite sure what is meant here, given that this information was already provided in detail in the section of the introduction that precedes the sentence marked by the reviewer.

Line 52:.. most recent literature findings… What does this encompass? Time aspect? What kind of search?

Reply: We have now specified this by adding the following sentences:

“We searched PubMed up to April, 2019, without language restrictions, with different combinations of the search terms “parenteral nutrition”, “PICU”, “early”, pediatric critical illness”. We focused on publications of the last 8 years, discussed in the context of earlier work.”

Timing of PN initiation:

Line 57-59: “Therefore, guidelines used to recommend that when provision of enteral nutrition (EN) is insufficient, impossible or contra-indicated, supplemental PN should be initiated soon after PICU admission (ref 7,12,13).

To my knowledge not any of these references have advocated PN soon after PICU admission, ref 7: In small preterm infants, PN must be instituted shortly after birth, in older children and in adolescence up to 7 days may be tolerated, depending on age, nutritional status, and the disease, surgery or medical intervention. Ref 12: No evidence for or against the need for nutritional support during the first week of life (Conventional feeding regarded as after 48 h). Ref. 13: Current practice is the initiation of enteral feeds within 48-72 h after admission to a PICU, and PN should only be considered if EN is not possible.

The authors need to find other references or rephrase.

Reply: Agreed and adapted as requested. We have now modified this sentence, since the recommendations on timing of initiation of supplemental PN in the referred older guidelines are indeed variable. However, because of this lack of clarity in the guidelines, a survey investigating nutritional practices in PICUs worldwide has been performed. This showed that PN was started within 48 hours after admission in the majority of the responding PICUs.

This sentence now reads as follows:

“Therefore, guidelines used to recommend that when provision of enteral nutrition (EN) is insufficient, impossible or contra-indicated, supplemental PN should be initiated.”

Line 69: early PN (please specify)

Reply: We have now specified early PN in this context. The sentence now reads as follows:

“In critically ill adults, the large multicenter EPaNIC (Early versus Late Parenteral Nutrition in ICU, n=4640) randomized controlled trial (RCT) showed that withholding of supplemental PN until day 8 of ICU stay (late PN), thus accepting a substantial macronutrient deficit, was associated with fewer ICU infections, a shorter duration of mechanical ventilation and renal-replacement therapy, and a shorter ICU and total hospital stay as compared with initiating supplemental PN early (within 48 hours after ICU admission).”

Since “early PN” in PEPaNIC was initiated within 24 hours, we have now also adapted this in the relevant section on the data of PEPaNIC:

“Withholding supplemental PN during the first week in critically ill children resulted in fewer new infections, a shorter dependency on mechanical ventilation and general intensive care, and a shorter hospital stay as compared with providing PN early, within 24 hours after PICU admission”.

Line 73-74: This is the same sentence as line 38, and same comment

Reply: As compared with critically ill adults, critically ill children both have limited macronutrient stores and relatively higher energy requirements. The limited stores are therefore not only due to the higher requirements. To clarify this distinction, we have now rephrased this sentence to:

“As compared with adults, critically ill children have limited stores of energy, fat and protein, and they also have relatively higher energy requirements.”

Line 106-109: This is also from the PEPaNIC trial and this could be clearer in the text

Reply: Adapted as requested. We have now clarified this. The sentence now reads as follows:

“A larger longitudinal study on all PEPaNIC patients with weight z scores available on admission and on the last day in PICU showed that weight deterioration during PICU stay was associated with worse clinical outcomes, but that withholding supplemental PN during the first week did not alter weight z score deterioration during PICU stay.”

Line 119-122: Only other trial sited. The authors do not discuss the differences between the findings in this trial and the findings in the PEPaNIC trial. This could be elaborated.

Reply: Adapted as requested. We have now modified this section by discussing all available studies on the subject, of the last 8 years.  This section now reads as follows:

“Apart from the PEPaNIC RCT, no other randomized controlled trial investigating the use or timing of supplemental PN in critically ill children has been published in the last 8 years. A limited number of observational studies on the use of supplemental PN and over- and underfeeding in PICU showed different results [25,28]. A retrospective single center study showed that late initiation of supplemental PN was associated with a higher nosocomial infection rate as compared with early initiation of supplemental PN [28]. In contrast, an observational study in 31 PICUs showed that the use of PN in general was associated with higher mortality [2]. Another retrospective study determining the incidence of over- and under-feeding in 139 children admitted to a tertiary PICU, showed that underfeeding was associated with shorter duration of PICU and hospital stay, as well as with fewer ventilation days, as compared with appropriately fed and overfed patients [25]. However, the observational design of these studies holds risk of bias by confounding variables, especially in nutritional research [29]. Therefore, comparison with the results of the PEPaNIC RCT is challenging. Further randomized controlled trials are warranted to determine the ideal time point for initiation of supplemental PN in the PICU.”

PN and the role of macronutrients

Line 159 and 160: The possible explanation seems a bit odd. Most observational studies a less stringent than RCT, thus with more diversity. RCT are defined, and will often have less differences between the included subjects. I do not know if it is correct to say that it was the RCT design, but rather the lack of restrictions on the participating centers, so that the nutritional intake (apart from when to start PN), had an observational character…

Reply: We do not agree with this statement. We underscribe the superiority of randomized controlled trials regarding their level of evidence as compared with observational studies. For the investigation of the impact of nutritional management in particular, observational studies are highly susceptible to bias introduced by confounding variables, most importantly by severity of illness. An RCT is the most reliable method to avoid such bias. In addition, the PEPaNIC RCT did not exclude important diagnostic categories.

Impact of PN on long-term outcome

According to this review, parents reported a better inhibitory control among children in the late PN group of the PEPaNIC trial after 2 years.

Reply: Agreed. This is the most important finding of the 2-year follow-up study of the PEPaNIC RCT.

Line 200-204 “Delaying supplemental PN has important consequences for daily life and the social environment later in life” …..  and further … “ supports the hypothesis that the harm induced by early supplemental PN might be caused by a direct metabolic insult….”

The authors are very conclusive. These limitations of the PEPaNIC trial does not allow such firm conclusions.

Reply: We agree that these findings need to be confirmed by future RCTs and that the underlying mechanism of this hypothesis needs to be unraveled. We have now reformulated this section, which now reads as follows:

“Since poor inhibitory control in children contributes to impulsive and destructive behaviors that upset or harm others [57], delaying supplemental PN can have important consequences for daily life and the social environment later in life. The long-term effects of late versus early supplemental PN were more pronounced in patients who were younger than 1 year of age at the time of PICU admission as compared with older children. This age-dependent vulnerability supports the hypothesis that the harm induced by early supplemental PN might be caused by a direct metabolic insult on the developing brain, since it was not statistically explained by the acute effects of the intervention itself, such as the increased incidence of new infections or delayed recovery. However, further research is warranted to unravel underlying mechanisms that would provide support for this hypothesis.”

Conclusions and newest guidelines

Line 287: …”compared with initiating supplemental PN early”, please specify

Reply: Adapted as requested. This sentence now reads as follows:

“A large multicenter RCT showed that withholding supplemental PN throughout the first week in PICU was clinically superior for the short-term outcome as compared with initiating supplemental PN within 24 hours after admission [19], independent of age [19] and nutritional status [21].”

Include a paragraph on the lack of studies found for this review and the limitations for the conclusions.

Reply: We have now rephrased the end of this section, which now reads:

However, the lack of other RCTs in this specific field makes it challenging to compare these findings with other available recent studies. Further research, in the form of multicenter RCTs, is warranted to determine the optimal composition and ideal timing of initiation of supplemental PN in critically ill children.”

Line 295- 297: “Further research, in the form of multicenter RCTs, is warranted to determine the optimal composition and timing supplemental PN beyond the first week of critical illness in children.

Please rephrase, we still do not know if PN started at 3, 4, 5 or 6 days are inferior to withholding PN for a week.

Reply: Adapted as requested. This sentence now reads as follows:

“Further research, in the form of multicenter RCTs, is warranted to determine the optimal composition and ideal timing of initiation of supplemental PN in critically ill children.”

Round  2

Reviewer 1 Report

The paper was improved but the authors still cite unpublished data. It should not take place in the scientific papers. If this data is so important the authors should wait for their publication and then use them.

Reviewer 2 Report

Thank you for the revision of the manuscript, which I think has improved the work.

I have no further comments.